# Titanium Implant Surface Effects on Adherent Macrophage Phenotype: A Systematic Review

**DOI:** 10.3390/ma15207314

**Published:** 2022-10-19

**Authors:** Manju Pitchai, Deepak Ipe, Santosh Tadakamadla, Stephen Hamlet

**Affiliations:** School of Medicine and Dentistry, Griffith University, Gold Coast Campus, Southport, QLD 4222, Australia

**Keywords:** titanium, macrophage, phenotype, implant, topography, hydrophilicity, osseointegration

## Abstract

Immunomodulatory biomaterials have the potential to stimulate an immune response able to promote constructive and functional tissue remodeling responses as opposed to persistent inflammation and scar tissue formation. As such, the controlled activation of macrophages and modulation of their phenotype through implant surface modification has emerged as a key therapeutic strategy. Methods: Online databases were searched for in vitro studies between January 1991 and June 2020 which examined the effect of titanium implant surface topography on the adherent macrophage phenotype at either the gene or protein level. Results: Thirty-nine studies were subsequently included for review. Although there was significant heterogeneity between studies, treatment of titanium surfaces increased the surface roughness or hydrophilicity, and hence increased macrophage attachment but decreased cell spreading. Physical coating of the titanium surface also tended to promote the formation of cell clusters. Titanium and titanium-zirconium alloy with a micro- or nano-scale rough topography combined with a hydrophilic surface chemistry were the most effective surfaces for inducing an anti-inflammatory phenotype in adherent macrophages, as indicated by significant changes in cytokine gene expression and or cytokine secretion profiles. Conclusions: The published data support the hypothesis that incorporation of specific topographical and physiochemical surface modifications to titanium can modulate the phenotypic response of adherent macrophages.

## 1. Introduction

Implants manufactured from titanium are a well-established treatment modality for the anchorage of prosthetic devices into bone in a process known as osseointegration. Titanium has been the material of choice for many medical devices such as hip, knee and dental implants due to its excellent mechanical properties and high degree of biocompatibility [1]. Biomaterials such as titanium however when implanted into the body, triggers a host immune response. In this regard, the early in vivo work by Donath and colleagues (1984), was the first to clearly demonstrate the importance of macrophages in this response and their subsequent role in peri-implant endosseous healing [2]. Macrophages, which are key early mediators of the host response, are a phenotypically heterogeneous population which following their arrival at sites of inflammation, become activated in response to signals present in the tissue to increase their production of cytokines, chemokines, and other molecules that contribute to the local inflammatory response.

Immune system biology has shown that within injured tissue including bone following implant placement, immune response mediators such as macrophages polarize into different phenotypes depending upon the signals received during their activation. These signals arise from their interaction with the titanium surface and thus differential macrophage responses may be critical in the overall osseointegration process given macrophages play dual roles as a major modulator of the initial healing response, as well as in the formation of osteoclasts involved in the later remodeling phase of bone homeostasis [3]. Biomaterial-induced modulation of macrophage function, phenotype and polarization to varying topography, has been a subject now of intense research for several decades [4] (for review). In this regard, Arron and Choi (2000) coined the term “osteoimmunology” to describe this interdisciplinary research field, that concentrates on the potential interplay between the skeletal and immune systems [5]. Moreover, osteoimmunology provides a heuristic concept to explore effective novel therapies for bone defect repair and regeneration [6,7].

The phenotype of macrophages present at the implant site can be broadly defined by their functional properties and are generally referred to as having either a M1 or an M2 phenotype mimicking the Th1/Th2 nomenclature described for T-helper cells [8]. The M1 macrophage phenotype is characterized by the expression of high levels of pro-inflammatory cytokines, high production of reactive nitrogen and oxygen intermediates, promotion of a Th1 response, and strong microbicidal and tumoricidal activity. In contrast, M2 macrophages are characterized by low levels of pro-inflammatory cytokines and high expression of anti-inflammatory cytokines which play a major role in promoting growth and regeneration. M2 macrophages have been further sub-classified (M2a, b, c and d), based on the type of stimulation and the subsequent expression of surface molecules and cytokines which reflect functional and molecular specializations [9].

For an implant to become osseointegrated, it therefore needs to trigger an overall osteo-formative response around the titanium surface at both cortical and cancellous bone levels. The interplay between M1 and M2 dominated microenvironments and the temporal modulation of the transition from M1 to M2 driven by interaction with the implant surface, may therefore be critical in determining the implant tissue response and ultimately the fate of the implant through the release of anti-inflammatory and proinflammatory cytokines, respectively.

Despite its superior mechanical properties compared to other biomaterials, titanium does not possess any osteoconductive or osteoinductive properties by itself. Hence, to try and improve the rate and or degree of bone formation around implants, significant research on the modification of titanium’s surface properties such as its topography (i.e., roughness) and physiochemistry (hydrophilicity and biofunctionalization with various polymers, peptides, etc) has been performed in this regard. For example, studies using topographically modified implants that aimed to reproduce the morphology of native bone, clearly showed that microtopography and or hydrophilic implants performed better clinically at influencing contact osteogenesis [10,11,12].

Following these attempts to further improve the rate and or degree of osseointegration, and thereby subsequently enhance the ultimate success rate of implants, there has been a paradigm shift in the development of titanium implants away from being classically inert, to being immunomodulatory, i.e., able to stimulate a host immune response that provides an osteogenesis-enhanced environment for bone producing cells. However, in vitro and in vivo studies using the same biomaterials have often been shown to yield varying results [13]. While significant progress has been made towards defining the systemic signaling pathways which underlie the different forms of macrophage activation, the influence of the titanium surface itself when in contact with macrophages has not yet been systematically studied. This study therefore aims to review the in vitro research data to determine whether surface modification of dental implant surfaces, promotes a regenerative associated phenotype in adherent macrophages.

## 2. Materials and Methods

### 2.1. Review Question

The review search question was formulated using the ‘PICO’ framework [14], with dental implants as the ‘Population (P)’ cohort, surface modification as the ‘Intervention (I)’, and macrophage phenotype as the ‘Outcome (O)’. No ‘Comparison (C)’ was defined. Hence, the formulated question was, “Does surface modification of dental implant surfaces, promote a regenerative macrophage phenotype?”. The PRISMA reporting guidelines [15] for systematic reviews were subsequently followed.

### 2.2. Search Strategy

Electronic searches of the PubMed, Ebsco, Embase, Scopus, Wiley Online and Ebsco Dent databases were performed. The search strategy used in PubMed, comprising free keywords and Medical Subject Headings is presented in Table 1. Manual search was conducted through screening the references of the included studies for identifying additional studies. All the experimental in vitro study designs were included. The search was limited to include studies published in English from 1 January 1991 to 1 June 2020. Conferences abstracts, letters to the editor, and studies that did not specifically assess the macrophage phenotype were excluded. Initial screening of the titles and abstracts was performed independently by one investigator (MP) and was reviewed by another investigator (ST). A list of potential papers was subsequently compiled, and the full text of these selected articles was reviewed to confirm their fulfilment of the inclusion criteria.

### 2.3. Inclusion and Exclusion Criteria

Articles were included for review if the studies considered the effect of titanium implant surfaces on macrophage phenotype at either the gene or protein level. Studies carried out entirely in vitro, or those in vivo studies which included some in vitro analysis, were included. Articles were excluded if implants other than dental implants were investigated, i.e., studies reporting on the effect of macrophages related to artificial joint replacement, plates for maxillofacial rehabilitation and other implants outside the oral cavity for purposes other than dental restoration. Studies on microorganisms which influence the regulation of the immune system and reactive lesions in patients that used dental implants except as a control group were also excluded. Review articles, letters, personal opinions, book chapters, conference abstracts and articles published in a language other than English were also excluded.

### 2.4. Data Extraction

All results retrieved from the searches were exported into reference management software and the duplicates removed. Those articles found to be appropriate through title and abstract screening were considered for further full-text review. Summary data from the selected full-text articles were extracted and split into six tables under the following subject headings: Reference, Author, Cell Type, Surface Modification, Surface Coating, Morphological Changes, Proliferation Changes, Osteogenic Activity, Gene and Cytokine Expression Changes and Study Conclusion(s). These subject heading were selected as the most consistent across the selected articles that broadly described (1) the methodology utilised (Cell Type, Surface Modification, Surface Coating); and (2) the macrophage phenotypic changes observed (Morphological Changes, Proliferation Changes, Osteogenic Activity, Gene and Cytokine Expression Changes). A glossary of abbreviations used is provided as Appendix A. Whilst a formal meta-analysis of the variables (described by subject headings) could not be performed due to the significant heterogeneity noted in the study methodologies and data reporting systems, where possible, a semiquantitative analysis was attempted.

### 2.5. Risk of Bias

To avoid any risk of bias in manuscript selection process, the ‘ToxRTOOL Ver. 1.0’ assessment tool was used to assess the data provided [16]. This tool evaluates the quality of toxicological data from in vitro studies by assigning a binary score of 1 or 0 for eighteen different ‘reliability criterion’. These criteria are then categorised with an overall ‘Klimisch category’ [17] of 1 (reliable without restrictions), 2 (reliable with restrictions) or 3 (not reliable) following simple addition. Category 1 is assigned for scores ≥ 15, category 2 for scores 11–14 and category 3 for scores < 11.

## 3. Results

### 3.1. Data Retrieval

A flow chart demonstrating the records retrieved from the database searches, included and excluded based on eligibility criteria is presented in Figure 1. Initial screening of the databases yielded a total of 606 articles. After elimination of duplicate records (108), screening of titles and abstracts resulted in 53 relevant studies were included for full-text review. Thirty-nine of these studies [13,18,19,20,21,22,23,24,25,26,27,28,29,30,31,32,33,34,35,36,37,38,39,40,41,42,43,44,45,46,47,48,49,50,51,52,53,54,55] which only assessed in vitro investigations were included in this systematic review.

### 3.2. Surface Topography

The most studied biomaterial was commercially available titanium alloy (Ti-6Al-4V) used in either a disc or screw shape of various diameters and dimensions with a range of surface modifications (Table 2). The surface modifications could be broadly grouped according to the methodology used in their production, i.e., (1) subtractive techniques (16 studies), e.g., sandblasting, acid etching; (2) additive techniques (11 studies), e.g., plasma spraying, nanotube formation, or (3) coating techniques (17 studies), e.g., covering the native titanium surface with inorganic or organic polymers. Of the subtractive techniques, a combination of sandblasting and acid etching were the most frequently used surface modifications [18,19,20,21,22,23,24,25,26,27,28,30,31,32] producing a hydrophobic microrough surface. Of these, five studies (36%) subsequently induced super-hydrophilicity to the already sandblasted and acid etched surface [19,21,23,28,31]. Of the additive techniques, the induction of nanoscale roughness or nanotube formation (internal diameters from 30–100 nm) by anodization at 5–100 volts were the most prevalent [13,22,33,34,35,36,37,38,39]. Micro-arc oxidation was used in a further two studies [40,41] to produce a micro-rough surface. Four studies included both subtractive or additive and coated surfaces for comparison [25,27,32,42]. Inorganic osteogenic materials, e.g., hydroxy apatite, chitosan and zirconia were the most frequently used coating materials [25,27,32,42,43,44,45,46,47,48,49,50,51], while a smaller number of studies examined the effects of natural compounds such as bacterial lipopolysaccharide (LPS) [52], interleukin-4 (IL-4) [53], silk fibroin [54] and collagen [55].

Macrophages studied included both primary human (7 studies) and rodent cells (5 studies) and those from human (THP-1, 2 studies) and rodent monocytic cell lines, i.e., J774.A1 (3 studies) and RAW 264.7 (22 studies) (Table 2).

### 3.3. Cell Morphology

Cell shape changes have often been associated with different functional states of cells. McWhorter (2013) demonstrated that macrophages polarized toward different phenotypes in vitro also exhibited dramatic changes in cell shape, with M2 cells exhibiting an elongated shape compared with M1 cells [56]. As such, striking morphological differences in macrophages were also reported in many of the included studies cultured on the various surfaces (Table 3 and Table 4). Although not examined in a consistent manner, some studies reviewed showed oval shaped M1 associated cells were often seen on polished titanium surfaces, e.g., studies [13,23,38], whereas surface modification to include plasma spraying [13] and/or the addition of nanoscale features [13,26,34,38] was associated with significant cell spreading and elongation. Indeed, the study by Pan (2017) showed that the further addition of nanofeatures to an already plasma sprayed surface resulted in multi-directional elongation and spreading and 3D distribution of the cytoskeleton [13].

Overall, of the 39 studies selected in this review, 25 studies (64%) also reported morphological changes in cells that were attached to the test (modified) surface when compared to the control surface (Table 3). To help assess the possible significance of surface driven morphological changes, the various surface topographies used these studies have been broadly grouped as either ‘smooth’ (Ra < 100 nm) or ‘rough’ (blasted, etched etc where Ra > 100 nm) accordingly. 96% of the studies on a rough surface described morphological changes such as ‘increased cytoplasmic volume’, ‘granularity’ and ‘extension of pseudopodia’ following culture compared to only 15% in macrophages cultured on a smooth titanium surface. Three or 11% of studies described morphological changes such as the cell shape becoming more ‘spindle-shaped’ or ‘elongated’, developing ‘pseudopodia extensions’ or ‘cell-spreading’ or ‘fully spread lamellipodia interacting with the surfaces’ on both smooth and rough titanium surfaces [47,53,55]. Only one study (2.5%) however showed the changes in macrophage morphology were more prominent on the smooth compared to rough titanium surfaces [23].

Subsequent statistical analysis clearly supports the proposal that a rough titanium implant topography promotes the activation of an M2 macrophage phenotype in adherent cells (chi-square statistic with Yates correction 29.301, *p*-value < 0.001).

### 3.4. Cellular Response

Regardless of the type of surface modification fabricated on the titanium implants, almost all studies included in this review reported significant immunological responses by the macrophages (Table 5). Of the 39 studies selected, only 5% of studies did not report any measure of cytokine expression by macrophages [18,48]. In the remaining 37 studies, inflammatory cytokine gene expression (as measured by either fold change or relative expression levels) or secreted cytokine levels (measured by immunoassay) were upregulated on rough titanium surfaces compared to the smooth surfaces. E.g., the microrough surface topography produced by combined sand blasting with acid etching of the titanium surface [19,21,23,24,25,31,32], was shown to induce the upregulation of pro-inflammatory cytokine gene expression, or cytokine secretion of IL-1β, IL-6 and TNF-α in adherent macrophages. Further modification of rough surfaces to induce super-hydrophilicity [18,19,21,23,28,30,31,32,39] showed this modification could induce a switch in macrophage phenotype from pro-inflammatory to a regenerative ‘M2-like’ phenotype, i.e., upregulation of IL-4 and IL-10 expression and concurrent down-regulation of the pro-inflammatory markers.

These studies (Table 5) suggest this surface modification, i.e., hydrophilicity, could potentially promote faster wound healing during osseointegration. In vitro evidence to support this hypothesis was clearly demonstrated in those studies [28,31,32] in which the osteogenic effects of this titanium surface-derived macrophage phenotypic changes on osteoblasts were examined using co-culture or conditioned media studies, i.e., the highest levels of mesenchymal stem cell (MSC) recruitment were seen with either M2 activated macrophages or macrophages on rough hydrophilic titanium [28]. Osteoblasts co-cultured with macrophages on modSLA titanium surfaces resulted in >2-fold increases in the expression of TGFβ / BMP signalling genes [31], and increased expression of RunX2, Sp7, Bglap, Alp, Bmp2 and Vegf was seen in co-cultured MSC’s [32].

Similar results of significant paracrine osteogenic effects in osteoblasts or osteoblast progenitors (Table 5) were also shown in many of the included studies [22,27,32,33,34,35,36,37,38,39,40,41,43], whereby the titanium surface modification resulted in a shift of macrophage phenotype (M1 to M2), e.g., increased β-catenin and osterix expression [27] and increased TGFβ1, BMP2, VEGF and decreased TRAP expression [40].

Further studies carried out on titanium surfaces modified at the nanoscale level showed that incorporating nanotubes or grooves onto the surface could also result in a favourable osteo-immunomodulatory microenvironment [26,34,35,36,37,38]. The importance of the physical dimensions of the nanotubes was further shown to be an important factor on the behaviour of macrophages whereby smaller diameter nanotubes were associated with an M1 macrophage phenotype characterized by high levels of secreted IL-1β, TNF-α and iNOS compared to larger diameter nanotubes which induced an anti-inflammatory M2-like macrophage phenotype in adherent macrophages with enhanced IL-10 and Arg-1 gene expression [34,35,36,37]. These results agree with those by Chamberlain where 70 nm diameter was demonstrated to be optimal for minimizing the macrophage inflammatory response [57]. In a further nanotube modified surface that also incorporated zinc [38], adherent macrophages showed enhanced gene and protein expression of the M2 markers TGF-β and heme oxygenase-1, whereas M1 markers TNF-α and IL-6 were moderately inhibited thus establishing an osteogenic microenvironment that would be conducive for bone formation. Similarly, other nanoscale modified titanium surfaces produced by etching [22], plasma-spraying [13] or nano-wire addition [33,39] also promoted immunomodulatory behavior that would favor osteogenesis and angiogenesis.

Of the studies using a surface coating on titanium to support the migration, proliferation and differentiation of macrophages and reduction in the inflammatory response, the application of hydroxyapatite significantly increased macrophage adhesion and downregulated pro-inflammatory mediators [43,44,46,47,49]. Similarly, various other biomaterial coatings, e.g., Bioglass [44], silk sericin [45], LPS [52], Chitosan [48], IL-4 [53], Zinc [42], GPTMS [50], wound-healing peptides [54], aspirin [55] and anti-osteoporosis drugs [51], showed functionalization of the surface while maintaining the underlying topography (Table 5). This could also significantly increase the initial attachment of immune cells and alter the immune cells response, although this result was dependent upon the biomimetic agent(s) used.

As for native titanium, hydrophilic modification of a zirconia-titanium alloy (RXD -Roxolid^®^, Straumann, Basel, Switzerland), similarly downregulated pro-inflammatory cytokines IL-1β and IL-6, producing an anti-inflammatory microenvironment by inducing macrophage activation similar to the anti-inflammatory M2-like state and increased levels of the cytokines IL-4 and IL-10 [23,25,28]. In fact, this hydrophilic implant (RXD-SLActive), when compared to other immune-modulatory titanium surfaces induced the highest level of osteogenic factor released from MSC’s and anti-inflammatory factors from macrophages with the lowest level of pro-inflammatory factors [32].

### 3.5. Risk of Bias

In this review, the thirty-nine studies were all found to have reliability scores of ≥15 (Klimisch category 1) indicating that data from these studies is reliable without any restrictions (Appendix A). None of the studies had any unreliable data (category 3).

## 4. Discussion

The findings of the studies (Table 6) included in this systematic review, clearly support the hypothesis that incorporation of titanium surface modifications increasing surface roughness and hydrophilicity with or without additional application of bioactive coatings, can promote a regenerative or M2-like phenotype in adherent macrophages which may then have the potential to enhance osteogenesis in BMSC’s in a paracrine manner.

More specifically, titanium surface roughness is well known to increase the surface area of implants and ultimately enhance osseointegration when compared with smooth surfaced implants, however further modification of this rough surface to increase surface energy thus promoting super-hydrophilicity, not only down-regulates the initial pro-inflammatory response by macrophages, but up-regulates an anti-inflammatory phenotype able to further promote wound healing. Topography-directed macrophage polarization is therefore a biologically feasible mechanism to assist in the design of implant surfaces aimed at promoting osteogenesis and osseointegration. Unfortunately, this review of in vitro studies only does not allow any determination of whether an amelioration of inflammation or promotion of anti-inflammatory mediators may be responsible for improved osseointegration seen clinically with specific surface modified titanium implants.

Of the 39 papers reviewed in this study, only one, Morra et al. (2015), provided data that suggested some caution should be used when assessing the potential impact of surface modification on the subsequent crosstalk between cells of the immune and skeletal systems [52]. These authors showed commercially available dental implants induced variable levels of expression of endotoxin-stimulated pro-inflammatory genes such as IL-1, IL-6, TNF-α, MCP-1, COX-2, and MCSF in murine J774-A1 macrophages, sometimes above the level expected to promote bone resorption in vivo. Moreover, the results were unaffected by the specific surface treatment; rather, they more likely reflected the level of care in the cleaning and packaging protocols of the manufacturers. Evaluation of adherent endotoxin should therefore be reappraised and considered amongst the relevant surface properties of implantable biomaterials for proper understanding of the tissue response to implants.

The mechanism through which the presence of a rough, hydrophilic surface topography and chemistry affects the osteoimmune response of macrophages in comparison to other distinct topographies is not yet understood. Li et al. (2020) showed higher hydrophilicity and surface free energy in anisotropic nanowire-like textured titanium promoted the availability of RGD binding domains in fibronectin and fibrinogen adsorbed onto the titanium surface [39]. These tripepetide Arg-Gly-Asp (RGD) domains provided more α5β1-integrin-specific instructions to MSCs, enhancing cell spreading and osteogenic differentiation. Furthermore, the authors suggested the combination of integrin α5-induced cell spreading and suppression of the interaction between fibrinogen and the integrin αM subunit, could act synergistically to cause accumulation of M2 macrophages on the nanowire-like textured surface.

In a subsequent coculture model, MSCs on the nanostructured surface exerted greater effects on naïve and M1 macrophages, causing them to adopt a less inflammatory macrophage profile characterized by reduced expression of IL-6 and TNF-α and concurrent increased expression of IL-10 and Arg1.

Alternatively, Pan et al. (2017) suggested the immunomodulatory properties of a plasma sprayed nanotextured surface, whereby macrophages were found to switch to M2 phenotype with decreased levels of inflammatory gene expression as well as increased expression of anti-inflammatory genes, were probably regulated by the ‘decisive’ role of cytoskeleton tension induced by specific cell shape when macrophages were cultured on this surface [13].

Regardless of the precise mechanism(s) responsible, biomaterial surface cues from immuno-modulatory surfaces interpreted by macrophages, results in the secretion of distinct cytokine profiles that are able to modulate osteogenic gene expression in osteoblasts in a paracrine fashion. Previous studies by our group have shown this may occur as a result of upregulation of the TGF-β/BMP signalling pathway [31]. Sun et al. (2013) also showed that TiO_2_ nanotube layers could stimulate RAW 264.7 macrophages to secrete BMP-2 in contrast to smooth surfaces [58]. Increasing nanostructure tube diameter further stimulated BMP-2 secretion. Further mechanistic studies by Li et al. (2020) also demonstrated that BMSC’s cultured on plasma sprayed nanotextured titanium mediated this immunomodulation via a ROCK-medicated COX2 pathway to enhance PGE2 production, which in turn acted on macrophages through the EP4 receptor and partially abrogated the activation of proinflammatory factors, specifically IL-6 [39].

An unfortunate limitation of this study was that a meta-analysis of the included papers was not able to be performed due to the significant heterogeneity found in the methods and outcome data presented by the study authors. Semiquantitative analysis where possible however (Table 3), did support in the affirmative that surface modification of dental implant surfaces could promote a regenerative macrophage phenotype as proposed in the research question. Lack of data on the physicochemical and mechanical properties of the titanium used in the included studies also presented as a major limitation of this review. Similarly, while most studies included subsequent osteogenic analyses in osteoblasts using co-culture or conditioned media, few examined any macrophage driven effects on mineralisation. While this disparity in reported studies continues, a strategy that similar future systematic reviews without meta-analysis could follow in order to allow for later synthesis has been proposed [59]. These ‘SWiM’ (synthesis without meta-analysis) guidelines have been included as Appendix A.

Finally, to help establish the veracity of our PICO question without the complications of unknown or unaccounted for systemic effects, this review has focused only on data arising from in vitro studies. Given the positive outcome, further systematic assessment of appropriate in vivo studies is now required to delineate the role of the biomaterial surface on the modulation of macrophage phenotype on in vivo osteogenesis.

## 5. Conclusions

In attempts to try and mimic the native tissue microenvironment, surface-modified titanium has been shown to modulate the function of adherent macrophages. Whilst any implanted device will result in an initial inflammatory response, modification of the device’s physiochemical properties to make it hydrophilic, treatment to add nanotube structures to the surface, or the addition of bio-functional surface coatings such as hydroxyapatite may reduce this initial inflammatory response and up-regulate a more regenerative phenotype in adherent macrophages, as suggested by the selected papers reviewed in this study. In vivo studies are now required to determine if these various modifications of potential implant surfaces will facilitate an enhanced rate or degree of osteogenesis not only in healthy individuals but also in immune-compromised patients.

## Figures and Tables

**Figure 1 materials-15-07314-f001:**
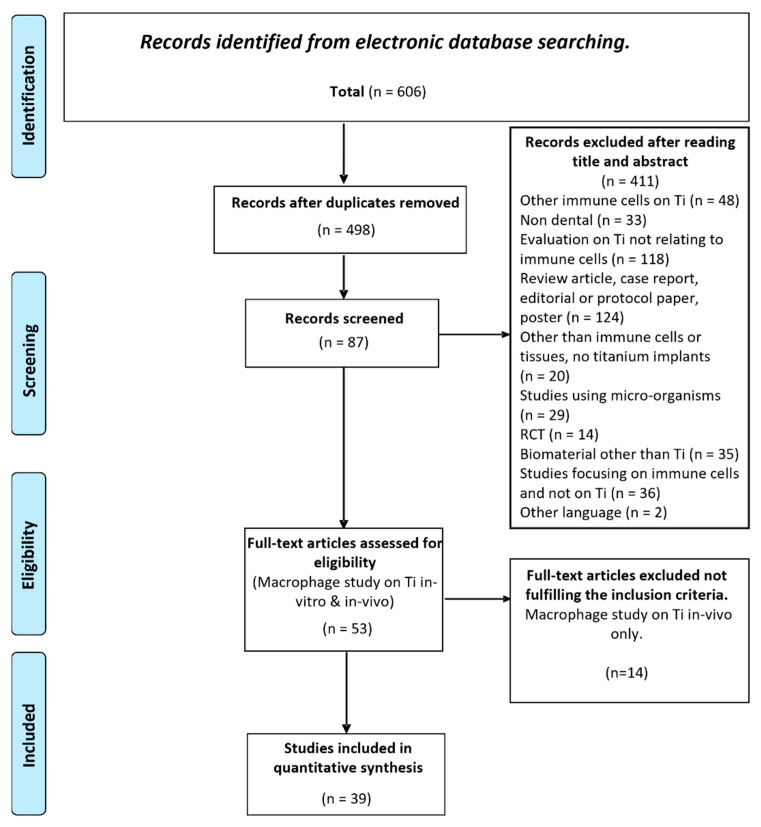
PRISMA flowchart of the screening process for the selected databases.

**Table 1 materials-15-07314-t001:** Search strategy used. PubMed database, “mh” represents MeSH terms and “all” represents search in all fields.

	Search String
1	“Titanium” (all)
2	“Dental implants” (all)
3	“Immune mediators” (all) OR “Macrophages” (all) OR “Neutrophils” (all) OR “Platelets” (all) OR “Lymphocytes” (all) OR “Cytokines” OR “Complement system” OR “proteins”
4	“Osseointegration” (mh)
5	#1 AND #2 AND #3 AND #4

**Table 2 materials-15-07314-t002:** Summary data (Cell type, Surface Modification and Surface Coating) of studies included in review. Both primary bone-derived macrophages as well as commercial cell-lines were included for analysis. Given the articles were selected based on the titanium eliciting an immune (macrophage) response, surface modification techniques were relatively evenly split between subtractive (41%) additive (28%) and coated (44%). Note that some studies included more than one methodology. (NA) not applicable in this study, (NR) not recorded in this study.

Ref.	Cell Type	Surface Modification(s)	Surface Coating
[13]	RAW 264.7	Polished (PT); Plasma sprayed (TPS); Nano plasma sprayed (NTPS)	NA
[18]	Human monocytes	Blasted and acid etched	NA
[19]	RAW 264.7	Polished; Blasted and acid etched (SLA); Hydrophilic SLA (modSLA)	NA
[20]	RAW 264.7	Polished; Blasted and acid etched	NA
[21]	THP-1	Polished; Blasted and acid etched; Hydrophilic, blasted and acid etched	NA
[22]	Rodent bone marrow macrophages	Smooth; Microrough; Nanorough	NA
[23]	Murine macrophage	Polished; Oxygen plasma treated; Blasted and Etched; Hydrophilic Blasted and Etched	NA
[24]	Bone marrow macrophages	Machined; Blasted; Blasted and acid etched	NA
[25]	Murine macrophages	Blasted and acid etched; Coated	Zirconia
[26]	RAW264.7	Polished; Blasted and acid etched; Physical grooves	NA
[27]	J774.A1	Coated; Blasted and acid etched	Strontium (Sr)
[28]	Primary macrophages	Smooth; Rough; Rough-hydrophilic	NA
[29]	RAW264.7	Coated	IL4 and Genipin hydrogel
[30]	Human whole blood	Blasted and acid etched; Plasma treated	NA
[31]	THP-1 and Rodent macrophages	SLA; modSLA	NA
[32]	Murine macrophages	Blasted and acid etched; Coated	Calcium phosphate
[33]	RAW 264.7	SLA; Nanowire; Nanowire +Zn	NA
[34]	Human monocytes	Polished; Nanotubes (NT5, NT20)	NA
[35]	RAW 264.7	Nanotubes (NT10, NT20)	NA
[36]	Murine bone marrow macrophages	Polished; Nanotubes	NA
[37]	Human monocytes	Polished (P); Nanotubes (NT5, NT20)	NA
[38]	RAW 264.7	Zn-incorporated TiO_2_ nanotube	NA
[39]	RAW 264.7	Nanowires (NW); Nanonests (NN); Nanoflakes (NF)	NA
[40]	RAW 264.7	Micro-arc oxidation (MAO); Steam hydrothermal treatment	NA
[41]	RAW 264.7	Micro-arc oxidation.	NA
[42]	RAW264.7	Micro-arc oxidation (MAO); Coated	Zinc acetate
[43]	J774.A1	Polished; Coated	Hydroxyapatite (HA)
[44]	RAW 264.7	Coated	Hydroxyapatite BioGlass Calcium Silicate
[45]	RAW 264.7	Coated	Serisin Sericin-RGD peptide
[46]	RAW 264.7	Coated	Strontium-Mg-Silicone Hydroxyapatite
[47]	RAW 264.7	Coated	HA-Clinoenstatite
[48]	RAW264.7	Coated	Chitosan Calcitonin bone morphogenetic protein 2
[49]	Human mononuclear cells	Coated	amorphous HA crystalline HA
[50]	RAW 264.7	Coated	3-glycidoxypropyl-trimethoxysilane (GPTMS)
[51]	RAW 264.7	Coated	Chitosan-β-cyclodextrin Calcitonin
[52]	J774.A1	Blasted and acid etched; Coated	LPS endotoxin
[53]	RAW 264.7	Low, Medium and High roughness	NA
[54]	RAW 264.7	Coated	peptide LL-37-loaded silk fibroin nanoparticle
[55]	RAW 264.7	Coated	Sodium hyaluronate and aspirin (ASA) nanoparticles

**Table 3 materials-15-07314-t003:** Morphological changes associated with an M2 macrophage phenotype were grouped according to whether this occurred on either a ‘smooth’ or ‘rough’ titanium surface. To facilitate this, given the multitude of ways the surfaces were prepared, a single roughness (Ra) value was used to group the data. M2 phenotype was significantly associated with attachment to the rough surfaces (chi-square statistic with Yates correction 29.301, *p*-value < 0.001).

Ref.	Author	SMOOTH (Ra > 100 nm)	ROUGH (Ra > 100 nm)
[13]	Pan 2017	NO	YES
[18]	Milleret 2011	NO	YES
[21]	Alfarsi 2014	NO	YES
[23]	Hotchkiss 2016	YES	NO
[26]	Kianoush 2017	NO	YES
[27]	Choi 2018	NO	YES
[29]	Yang2018	NO	YES
[32]	Hotchkiss 2019	NO	YES
[33]	Zhu 2019	NO	YES
[34]	Ma 2014	NO	YES
[35]	Wang 2017	NO	YES
[37]	Ma 2018	NO	YES
[38]	Chen 2020	NO	YES
[39]	Li 2020	NO	YES
[40]	Bai 2018	NO	YES
[41]	Bai 2018	NO	YES
[42]	Zhang 2018	NO	YES
[43]	Takebe 2007	NO	YES
[44]	Scislowska-Czarneck 2012	NO	YES
[47]	Wu 2015	YES	YES
[48]	Huang 2016	NO	YES
[49]	Rydén 2017	NO	YES
[53]	Zhang 2018	YES	YES
[54]	He 2019	NO	YES
[55]	Zhang 2019	YES	YES

**Table 4 materials-15-07314-t004:** Summary data (Morphological and Proliferative Changes) of studies included in review. (NA) not applicable in this study, (NR) not recorded in this study.

Ref.	Morphological Changes	Proliferation Changes
[13]	PT–Round shape. TPS–Elongated body. NTPS–multi-directional elongation and spreading and 3D distribution of the cytoskeleton.	Increased human umbilical vein endothelial cells proliferation when incubated with RAW 264.7 cell medium from NTPS surface.
[18]	Signifiicant amount of fibrous structures.	Macrophage/ monocyte number similar on both surfaces.
[19]	NR	Increased cell attachment at 24 h on modSLA surface.
[20]	NR	LPS/Interferon–50% lower cell number after 24 h on Po surface.
[21]	Pseudopodia-like extensions from the cell body	NA
[22]	NR	Micro < Nano surface at 3 and 7 d and > cells adherent to Nano surface at 14 d.
[23]	Macrophage cell number and morphology similar on smooth titanium surface and control surface whereas less elongated on rough surface. No giant cells.	NR
[24]	NR	Apoptosis assay–neither LPS nor titanium affected macrophage survival
[25]	NR	NR
[26]	More spread morphology on rougher grooved surface.	Novel grooved topographies altered cell morphology.
[27]	Sr-SLA–branched cell shape with lamellipodial projections and exhibited a larger cell size compared to SLA.	SrSLA displayed higher level of attachment and proliferation.
[28]	NR	Increased proliferation of activated T-cells treated with media from macrophages plated on titanium compared to plastic.
[29]	0.7% genipin +IL4–discoid shape. F-actin and pseudopodia appeared.	Higher viability on GG07-I and GG07 than on polished titanium.
[30]	NR	NR
[31]	NR	NR
[32]	Macrophages attached with little or no variation on the surfaces	NR
[33]	Zinc-decorated tianium surfaces inhibited the adhesion and proliferation of macrophages and induced M2 state.	Zn-decorated Ti surfaces inhibited the adhesion and proliferation of macrophages.
[34]	Polished and NT20–stretched spindle-like shape and no significant difference in the macrophage proportion. NT5–oval shape and enhanced cell spreading but inhibited cell stretching after attachment.	Fewer cells attached to NT5 surface than NT20 or polished surfaces.
[35]	NT20–Low density, appeared elongated and increased slowly from 1to 3 days.Titanium and NT10–stretched rapidly and exhibited a similar oval shape.	NT10 and NT20 reduced macrophage adhesion after 24 h. Minimum adherence on NT 20.
[36]	NR	NR
[37]	NT5 and NT20–cells well stretched. NT–aligned in a consistent direction. P–unordered distribution.	NT5 and 20 conditioned media increased proliferation at days 3 and 7.
[38]	Titanium–spherical and clustered morphology. Nanotubes–both round-like cells (M1) and elongated (M2) cells.	Release of Zn decreased cell activity and proliferation but did not increase cell apoptosis.
[39]	Ti and NF-Ti–native round morphology. NW-Ti and NN–Ti- multidirectional protrusions and elongated cells with spindle-like morphology.	Surface nanostructures alter cytoskeletal structures in macrophages, although these structures were relatively disorganized compared to those in MSCs.
[40]	MAO + steam–number of fully spread lamellipodia interacting with the surfaces. MAO–slight and planar lamellipodia.	MAO-H0.5 induced more robust macrophage adhesion and activation compared to other surfaces.
[41]	High temperature annealing enhanced macrophage activation. Considerable number of fully spread and rounded macrophages with greater filopodia.	Macrophage adhesion increased on high temp annealed surfaces (MAO-650).
[42]	Day 1–MAO showed more membrane protrusion and larger spreading area than Zn coated. Day 3 and 5–MAO and Zn coatings aggregated into clusters, exhibited spherical morphology.	Fewer cells on MAO c.f. heat-treated surfcae at day1. No difference at days 3 and 5.
[43]	Polished–spherical with numerous microvilli HA–more spread out and were characterized by a thin cytoplasmic rim with numerous microvilli adhering to the disk surface. At 72 h–extensive cellular networks.	No significant effect of HA
[44]	BG–Less flattened and no clusters. HA and BG–flattened cells and formed clusters.	BG–increased day 3 and 7 CS–reduction both days. HA–decreased day 7
[45]	NR	NA
[46]	NR	NR
[47]	RAW cells grew equally well on both HA and clinoenstatite surfaces.	Phagocytosis higher on HA-coated surfaces
[48]	Greater proportion of cells on uncoated substrates were of larger size and multiple nuclei compared with those cells on coated titanium substrate.	RAW264.7 on different substrates in the presence of M-CSF and RANKL could induce the differentiation of macrophage cells into osteoclasts
[49]	More adherent cells on amorphous HA in comparison to crystalline HA.	Higher ratio of adherent cells demonstrated for the amorphous HA
[50]	NR	NR
[51]	NR	NR
[52]	NR	NR
[53]	Both large giant cells and smaller, undifferentiated macrophages were visible on all surfaces.	Higher cell numbers for rougher surfaces.
[54]	Normal cell morphology, viability and spread	More migration of cells in Ti-SF/LL-37 group.
[55]	Titanium ASA– larger polygonal morphology. Titanium–spindle shaped cells.	Effect of ASA on cell proliferation concentration dependent. Combination of COLI and ACS improved migration of MSC’s.

**Table 5 materials-15-07314-t005:** Summary data (Osteogenic activity and Gene and Cytokine Expression changes) of studies included in review. (NR) not recorded in this study.

Ref.	Osteogenic Activity	Gene and Cytokine Expression Changes
[13]	Increased BMSCs calcium deposition when incubated with RAW cell medium from NTPS surface.	NTPS surface: M1 (INOS, TNFα, IL6, IL1β and IFNγ) -lowest levels. M2 (ARG, IL4, IL10 and IL1ra) - highest values.
[18]	NR	NR
[19]	NR	modSLA–down regulation of pro-inflammatory cytokines (TNFα IL1α, IL1β, CCL2) and upregulation of anti-inflammatory cytokines (IL4, IL10, IL11 and IL13).
[20]	NR	SLA surfaces did not activate Arg-1 and NOS2 expression, but relative to polished surfaces MCP-1 and MIP-1α were upregulated after 5 days, whereas the secretion of the M1-associated chemokine IP-10 was lowered.
[21]	NR	Key pro-inflammatory mediators (CCL-1, 2, 3, 4, 18, 19 and 20, CXCL-1, 5, 8 and12, IL1b, TNF, CCR7, LTB and LTB4R) downregulated on the modSLA surface c.f. SLA at day 3.
[22]	ALP mRNA raised at 7d. Decreased OPG with increased surface roughness (S > M > N)	Expression of prominent osteoclast-promoting factors TNFα and MCSF increased by BMSCs cultured on both micro- and nano-scale titanium topographies.
[23]	NR	Smooth Ti induced inflammatory macrophage activation (IL1β, IL6, and TNFα). Hydrophilic rough titanium induced anti-inflammatory interleukins IL4 and IL10.
[24]	SLA particles increased size and total area of TRAP+ve cells	SB particles induced the most severe inflammatory response (increased IL1β, IL6 and TNFα). Particles from sandblasted/acid-etched discs induced a milder inflammatory response.
[25]	NR	Increase in IL1b, IL6 and TNFα on Ti and TiZr surfaces, decrease in IL1b and IL6 on TiZr modSLA compared with TCPS.
[26]	NR	Galectin-3 inhibitor (lactose) down-regulated M2 marker (mannose receptor) while M1 marker (iNOS) was up-regulated on smooth and rough surfaces.
[27]	β-catenin was increased in cells grown on the Sr-SLA surface at early time points (3 and 7 days, Figure 9A).	SrSLA increased M2 phenotype (arginase 1, MR and CD163).
[28]	Highest MSC recruitment with M2 activated and macrophages on rough hydrophilic Ti.	Macrophages on hydrophilic Ti consistently released higher levels of anti-inflammatory factors (IL4, IL10).
[29]	NR	IL4 loaded (GG07-I) induced phenotype switch from M1 to M2 at 7 days. Decreased IL1β, IL6, TNFα. Increased IL10 and TGFβ1.
[30]	NR	24 h gene expression of proinflammatory cytokines decreased in control group and increased in test groups.
[31]	>2-fold increase TGFß / BMP signalling in OB’s cocultured with macrophages on Ti surfaces (modSLA vs. SLA).	M1 on modSLA increased CD163, Arg1, BMP, TGFβ M2 on SLA increased INOS, IL1β.
[32]	rough hydrophilic RXD-SLActive surface increased RUNX2, SP7, BGLAP, ALP, BMP2 and VEGFA from MSCs.	Rough hydrophilic RXD-SLActive surface induced the highest level of anti-inflammatory factors from macrophages with the lowest level of pro-inflammatory factors. Osseospeed and TiUnite implants supported lower levels of osteogenesis and increased secretion of pro-inflammatory factors.
[33]	Osteogenic capacity of TiSLA, Ti-NW and Ti-NW-Zn all enhanced.	Macrophages on Ti-SLA, Ti-NW and Ti-NW-Zn tended to be M2 phenotype rather than M1 phenotype (IL6 no change, increased IL10). No difference in M1 polarization.
[34]	osteogenic activity NT5 > NT20 > P	Nanotube surfaces enhanced osteogenic gene expression. NT20 surface showed greater osteo-inductive effect compared to the P and NT5 surfaces at all time points.
[35]	Osteogenic capacity of MC3T3 in CM from NT 20 was enhanced (NT 20 > NT 10 ≈ cp Ti)	NT 20 induced anti-inflammatory M2 macrophage state with increased IL10 and ARG, while NT 10 was associated with M1 macrophage phenotype with increased IL1β, iNOS and TNFα.
[36]	BMSC osteogenic activity was NT100 > NT30 > P.	Increased expression of iNOS and IL6 on NT100 Increased expression of Arg1 and IL10 on NT30.
[37]	Increased bMSC ECM mineralization on NT5 surface. Increased multinuclear giant cell and osteoclast formation on NT20.	NT20 - Increased IFNg and IL1b secretion NT-5 - Increased TGFb. Both NT5 and NT20 samples inhibited IL8 secretion.
[38]	TNT groups increased transcription levels of osteogenic related genes.	M1 cells: 15VZn and 25VZn moderate inhibition of IL6 and TNFα. M2 cells: TGFβ and HO-1 showed positive promotion.
[39]	Macrophage CM effect on BMSC’s NW-Ti > NN-Ti > NF-Ti.	BMSCs on nanostructured surface exerted greater effects on M0 and M1 macrophages, causing them to adopt a less inflammatory macrophage profile characterized by reduced expression of IL6 and Tnfα and concurrent increased expression of IL10 and Arg1.
[40]	MAO-H0.5 increased TGFβ1, BMP2, VEGF and decreased TRAP. MAO-H3 and -H6 decreased TGFβ1, BMP2, VEGF and increased TRAP.	MAO-H0.5 downregulated M1 markers IL1b, IL6, IL18, CD11c and CD86 compared to MAO-H3 and -H6. Increased expression of M2 markers, IL10 and CD206 on MAO-H6.
[41]	Increased metabolic activity of OB’s grown on MAO-450 and -650.	MAO-650 downregulated IL6, IL1b and TNFα. Facilitated transition to M2 with increased expression of IL10, CD206, and CD163.
[42]	MHTZn - high ALP activity, more calcium nodules.	TNFα, IL6, IL4 and IL10 upregulated on MHTZn compared to the MAO.
[43]	BMP2 secretion at 24h in HAcpTi cultures	The ratio of BMP2 mRNA was higher on HAcpTi than on ScpTi after 24 h.
[44]	NR	HA–TNF decreased on day 3 and 7 while IL6 and IFN was increased on day 3. BG–IL12 and IL10 unchanged while decrease in secretion of TNF and MCP1 on both days of culture. IFN decreased on day 7. CS–enhanced production of IL6 and IL12.
[45]	NA	Ti-SS and Ti-SSRGD–low NO production. Pristine Ti – higher NO, TNFα and IL1b in comparison to Ti-SS and Ti-SS-RGD.
[46]	NR	SMS–IL1ra upregulated. IL1b, IL6 and OSM expression downregulated. Osteoclast activity genes (TRAP, CTSK, CA2, RANK and MMP9) all significantly downregulated.
[47]	NR.	CLT–downregulated inflammatory gene (IL1b, IL6, IFNg and OSM) gene expression. Increased anti-inflammatory IL-10 secretion. Expression of osteoclast activity genes (TRAP, CTSK, CA2, RANK and MMP9 were all significantly downregulated.
[48]	The osteoclast-like cells on TC4/LBL/CT and TC4/LBL/CT/BMP2 implants displayed much lower TRAP activity than those cellson bare TC4 or TC4/LBL.	NR
[49]	Similar TGFβ1 in all groups. No BMP2 detected in any group.	Higher TNFα with amorphous HA at 24 h.
[50]	No difference in ALP activity between the sol-gel materials.	GPTMS (100G) - Increase in TNFα and IL10.
[51]	Anti-osteoporotic biofunctionalized Ti increased BMP2, VEGF, decreased MCSF, TRAP.	RAW264.7 cells grown on biofunctionalized Ti showed superior M2 phenotypical differentiation efficiency, but lower MCF/TRAP gene expression levels.
[52]	NR	LPS endotoxin stimulated IL1, IL6, TNFa, MCP1, COX2, and MCSF overexpression. IL1 and IL6 expression significantly dampened by full endotoxin-removal. Macrophages express same level of IL transcripts after endotoxin removal, irrespective of surface roughness.
[53]	NR	TiLR had significantly higher RANK and MMP9 gene expression than TiMR or TiHR.
[54]	No significant difference in ALP expression with Ti-SF and Ti-SF/LL-37 compared to Ti.	Ti-SF/LL-37- highest expressions of TNFα, TGFβ1, IL1β, IL-6, CCR7 and iNOS compared to Ti and Ti-SF.
[55]	Low ASA concentration promoted ALP activity in BMSCs.	ASA decreased IL6, TNFα and NaNO_2_ induced by LPS after 12 h.

**Table 6 materials-15-07314-t006:** Conclusions of studies included in review.

Ref.	Author	Study Conclusion(s)
[13]	Pan 2017	Tension-mediated immunomodulatory properties with shift of M1 to M2 phenotypes enhancing osseointegration.
[18]	Milleret 2011	Alkali-treated SBA Ti surfaces perform better in terms of osseointegration, a continuous and structured layer of blood components on the blood-facing surface supports later tissue integration of an endosseous implant.
[19]	Hamlet 2012	Modulation of the inflammatory response may facilitate the enhanced bone wound healing and osseointegration observed clinically using implants with a microrough hydrophilic surface.
[20]	Barth 2013	Macrophages on the SLA surface adopted elements of an M2-like phenotype, suggesting that when implanted the SLA surfaces may enhance wound repair.
[21]	Alfarsi 2014	Hydrophilic titanium surface can modulate human macrophage pro-inflammatory cytokine gene expression and protein secretion.
[22]	Nagasawa 2015	Difference in surface topography altered BMSC phenotype and influenced BMM osteoclastogenesis.
[23]	Hotchkiss 2016	The combination of hydrophilicity and increased surface roughness interact synergistically to yield a microenvironment suitable for reduced healing times and increased osseointegration.
[24]	Eger 2017	Particles from sandblasted discs induced more osteolysis than those from sandblasted/acid-etched discs.
[25]	Hotchkiss 2017	Increase in surface energy reduced proinflammatory cytokines and increased anti-inflammatory cytokines.
[26]	Kianoush 2017	Skewing of phenotype suggests a role for galectin-3 in macrophage polarization towards the M2 phenotype.
[27]	Choi 2018	Sr-containing nanostructures favourably influence early immunoinflammatory macrophage cell functions and functionality of osteogenesis cells.
[28]	Hotchkiss 2018	First study to show the importance of macrophage response to surface modifications (roughness and hydrophilicity) of metallic biomaterials and modulation of the adaptive immune system.
[29]	Yang 2018	Delayed release of IL4 by GG07-I promoted shift to M2 shift in the simulated inflammatory microenvironment.
[30]	Becker 2019	Surface plasma treatment showed reduction in proinflammatory cytokines during initial contact with human whole blood.
[31]	Hamlet 2019	Used defined macrophage populations to show Ti adherent macrophages modulate their phenotype in response to biomaterial surface cues resulting in the secretion of distinct cytokine profiles able to stimulate osteogenic gene expression in osteoblasts via the TGFß/BMP signalling pathway.
[32]	Hotchkiss 2019	Evaluated differences in cell response to commercially available clinical implants—not all surface modification procedures generate the same cell response.
[33]	Zhu 2019	Ti-NW-Zn surfaces not only provided excellent corrosion resistance properties, but also inhibited the adhesion of macrophages.
[34]	Ma 2014	Dominant role of macrophage-related inflammation in bone healing around implants. Surface nanotopography can have an immune-regulating effect in support of the success of implants.
[35]	Wang 2017	Nanotubular TiO2 surfaces were demonstrated to regulate macrophage polarization. The largest nanotubular dimension surface (NT20) showed the least M1/M2 ratio and the least production of pro-inflammatory cytokines with highest expression levels of TGFb, PDGF and MMP9 which favours an osteo-immunomodulatory microenvironment.
[36]	Wang 2018	TiO_2_ NTs (80–100 nm) induced M1 while TiO_2_ NTs (30 nm) induced M2 phenotype.
[37]	Ma 2018	NT surface topography and respective CM acted together to promote osteogenic behavior of bMSCs. Both NT5- and NT20-CM induced similar improvements in the osteogenic behaviour related to bioactive factors secreted by monocytes/macrophages exposed to the different surfaces.
[38]	Chen 2020	Zn-loaded TNT surfaces with a suitable diameter (15VZn group) could stimulate cytokine release by macrophages to act on osteoblasts, thereby inducing the osteoclast/osteogenesis balance developing toward osteogenesis.
[39]	Li 2020	NW-Ti, which has higher hydrophilicity promoted the availability of binding domains on adsorbed Fn, provided more α5β1-integrin-specific instructions to BMSCs and was capable of enhancing spreading and osteogenic differentiation.The combination of integrin α5-induced cell spreading and suppression of the interaction between Fg and the integrin αM subunit may act synergistically to cause the accumulation of M2 macrophages on the NW-Ti surface.
[40]	Bai 2018	HA- nanoparticles (MAO-H0.5) downregulated inflammation (compared to nanorods) and stimulated osteogenic and angiogenic factors that promote favourable osteoimmune environment.
[41]	Bai 2018	Highlights the profound effect of surface physicochemical properties on regulation of osteo / angiogenesis and osteo-immunomodulation.
[42]	Zhang 2018	TiO_2_/ZnO regulated the polarization of M1 and M2 and enhanced osteogenesis, compared to TiO_2_.
[43]	Takebe 2007	Macrophages have the capacity to adhere to HA/cpTi endosseous implants and provide a source of osteoinductive cytokines that may play a key role in the process of osseointegration.
[44]	Scislowska-Czarneck 2012	Improved bioactivity of titanium was achieved by the application of the hydroxyapatite and bioglass layers.
[45]	Nayak 2013	Sericin immobilized titanium surfaces are potentially useful bioactive coated materials for titanium-based medical implants.
[46]	Wu 2014	SMS coatings switch the macrophage phenotype into M2. Osteoclastic activities were also inhibited by SMS coatings.
[47]	Wu 2015	CLT coatings released Mg and Si ions, and induced an immunomodulation more conducive for osseointegration, demonstrated by downregulation of pro-inflammatory cytokines, enhancement of osteogenesis, and inhibition of osteoclastogenesis.
[48]	Huang 2016	The results indicated that CT or/and BMP2 embedded multilayer structure was capable of inhibiting the osteoclasts differentiation The results indicated that CT and or BMP2 embedded multilayer structure capable of inhibiting osteoclast differentiation.
[49]	Rydén 2017	Thin HA coatings with similar micro-roughness but a different phase composition, nano-scale roughness and wettability associated with different monocyte responses.
[50]	Araujo-Gomes 2019	Inflammatory potential GPTMS concentration dependent. Greater adsorption of complement proteins can condition macrophage polarization.
[51]	Chen 2020	Anti-osteoporotic biofunctionalized Ti implant effectively regulates the biological functions of osteoclast, osteoblasts, and macrophages to promote remodeling and healing of damaged bone tissues.
[52]	Morra 2015	Expression of proinflammatory genes, IL1 and IL6 is directly and selectively related to the amount of adherent endotoxin, and it is largely independent from surface topography.
[53]	Zhang 2018	Results suggest surface roughness is an important factor in mediating osteoclast−material interactions which determine the osteogenic differentiation of osteoblast progenitor cells and hence the process of osseointegration.
[54]	He 2019	Peptide LL-37-loaded silk fibroin nanoparticles improved cell viability, recruitment and paracrine responses of macrophages.
[55]	Zhang 2019	Ti surface containing ASA not only supported the migration, proliferation and differentiation of BMSCs but also reduced the inflammatory response of macrophages compared with Ti discs without surface modification.

## Data Availability

Not applicable.

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
