# Peer review of "Titanium Implant Surface Effects on Adherent Macrophage Phenotype: A Systematic Review"

_materials, 2022, doi:10.3390/ma15207314_

Round 1

Reviewer 1 Report

The work of Pitchai et al. is a systematic review on the effect of titanium implant surface effects on the adherent macrophage phenotype. The work is submitted to a special issue, "Dental Implant Surfaces: Controlling Hard or Soft Tissue Response", and it fits the SI's scope. Revisions could include and tackle the following points:

1. More details with respect to the surface nanotopography are needed, both in text (ln 152-169). How was the surface topogography for all selected articles from Table 2 and Table 3, what are the nanotubes sizes, or grooves or the specific topography? Some additional information can be added for the readers not specialized in this field.

2. With respect to the inclusion and exclusion criteria the authors used for the articles, there are works from several groups evaluating the behaviour of macrophage cells (adhesion, proliferation, phenotype (M1, M2), pro-inflammatory cytokine gene expression or cytokine secretion, at a first glance, some crucial references are missing. Authors need to consider also more representative works. For example for just one type of nanostructure, namely the case of nanotube surfaces on Ti or Ti alloys (by electrochemical anodization), the works of Chamberlain, or Brammer or Necula/Cimpean (both more recent and older works) investigating the effect of tube diameter on macrophage cells, including phenotype and cytokine secretion are missing. While the works of these groups may not mention dental implants, they investigate all the key points the authors mention.

Author Response

1. Author response: Thank you for your comments. Given the theme of this journal issue ‘Dental Implant Surfaces: Controlling Hard or Soft Tissue Response’, our research question designed according to the PICO principles, necessarily focused on the immune response to titanium implants. As such, we selected articles which subsequently reported cellular responses to a wide range of surface modifications that including nanotubes. The table headings ‘Surface modification’ and ‘Surface coating’ described in the selected articles, were chosen to reflect the main contributing factors as influencing the observed cellular responses and the specific surface modification (such as nanotubes) therefore was not the subject of the review per se.

For reader clarity, additional information regarding nanotube topography has been added to the manuscript as follows on Pg.5: “Of the additive techniques, the induction of nanoscale roughness or nanotube formation (internal diameters from 30 – 100nm) by anodization at 5 – 100 volts were the most prevalent [22, 33, 34 – 40].”

2. Author response: The authors agree that using ‘implants’ as a selection criterion, has indeed limited the number of published articles included in the review. As mentioned above this was a deliberate decision by the authors given both the large existing literature on the effects of surface modification on macrophage phenotype, in an attempt to narrow this focus to surfaces demonstrated on implants to fit the theme of the journal issue. This limitation of the review has subsequently been highlighted in the discussion.

Also Pg.13: “These results agree with those by Chamberlain where 70nm diameter was demonstrated to be optimal for minimizing the macrophage inflammatory response [58]. “

Reviewer 2 Report

This paper is the product of a massive undertaking by these Authors, and their synthesis the result of much work. There are extensive studies on this topic that were available for review. The paper is well written and shows good adherence to reporting guidelines. Nevertheless, the absence of a meta-analysis is disappointing.

1. At this point, this review provides an extensive qualitative review of the literature, given these search parameters. The Tables could be made more specific in terms of providing descriptive statistics for the various comparisons, not just a statement of significance or not. In that regard, the direction of findings is always more informative, and no less efficient, than ‘different’.

2. But consider this. On the one hand, the Authors appear to be comfortable with their synthesis of data from studies with disparate independent and dependent variables, and with their ability to conclude that rough surfaces are better than smooth, and that hydrophilic surfaces makes the best rough surfaces. To do this, the Authors would appear to have implicit rules that allow them to standardize those different methodologies. On the other hand, they argue that those disparate methodologies preclude their conduct of a meta-analysis. It would seem that the rules used for implicit standardization could be made explicit. This would then allow for a meta-analysis and provide the reader with quantitative results. That is, I suggest that they scale the different study outcomes in a consistent way that would allow for meta-analysis of the major question, those effects on M2 macrophage response. This would, of course, be constrained by the rules developed for the standardization of the M2 response, but these rules would provide a more transparent approach than the one that is now employed.

3. If the Authors believe that the disparate study designs preclude a MA, they could forego their strong conclusions and recommend a reporting strategy that future studies could follow in order to allow for later synthesis. However, it does not seem acceptable to say that a MA is not possible and yet come to conclusions that really require a MA for support. To do this, the Authors might follow guidelines published in the BMJ regarding systematic reviews without meta-analysis https://doi.org/10.1136/bmj.l6890.

4. Please define NA and NR and explain to the reader how studies with these apparently missing outcomes are retained in the review.

Author Response

  1. Author response: Thank you for your comments. To support the conclusions made in the study regarding surface effects on cell morphology and response, where possible we’ve added descriptive statistics (new Tables 2 & 3) qualifying their effects on ‘Cell morphology’ and ‘Gene & Cytokine Expression changes’.
  2. Author response: The authors agree a meta-analysis would have significantly supported the conclusions initially made in the report. Unfortunately due to the heterogeneity of the results presented in the selected papers, the data was not amenable to such analysis. While the descriptive statistics now added may assist in this regard, this limitation has now been included in the discussion and conclusions on Pg.’s 8, 9, 12 & 20.

  3. Author response: Thank you for the suggestion. The authors agree that without a meta-analysis, the strong conclusions as initially presented were not supported by any quantitative criteria. The manuscript has been modified accordingly to moderate the conclusions.

    We have also included semiquantitative analysis where possible (table 3) with statistical analysis to support the conclusions. Pg.’s 8, 9.

    The ‘SWiM’ guidelines for systematic reviews without meta-analysis as suggested to assist any future studies or synthesis of the data have been included as supplementary table 2.

  4. Author response: Additional text to all table legends added “NA” not applicable in this study, “NR” not recorded.

Reviewer 3 Report

Thank you for the opportunity to review paper titled “Titanium implant surface effects on adherent macrophage phenotype: A systematic review.”

The idea of macrophages role during osteointegration and osteoimmunology is not new but interesting thus the most important thing in such a review is to present the available knowledge regarding this topic in the most comprehensive manner. It is surprising that papers of H. Jennissen  “A macrophage model of osseointegration” or C. Guder “Osteoimmunology….” or  M.R. Taubman  are not even mentioned in the introduction or discussion section not to mention the main expert on this topic professor K. Donath. Thus, in my opinion, the introduction should be better presented and needs correction. The literature is quite poor and the reason behind it might be wrong search strategy and review question. It seems that the topic should be more extensively studied by the authors before writing such a paper. Tables are too extensive and hard to read and have to be reorganized. In my opinion authors have chosen very comprehensive topic instead of focusing on some smaller parts so they could cover it completely. I appreciate and acknowledge their hard work and in my opinion this paper has some potential but not in this form.

Author Response

Author response: The authors thank the reviewer for his comments. As suggested, the existing literature on biomaterial immunomodulation is now quite large and may be more amenable to a more selective systematic review where accompanying possible meta-analysis as suggested by reviewer 2 would provide more rigor to the conclusions. Notwithstanding this, our research question was deliberately designed to focus on the immune response to titanium implants to allow a better fit with the theme of this journal edition. This combination of search terms (table 1) whilst the authors agree is not comprehensive enough for a broader review of osteoimmunology, we feel it does focus more specifically on our research question.

Additional information however has been added to both the Introduction, Discussion and References to reflect this and to draw attention to the limitations of the study. The introduction has also been restructured significantly and the two large tables have been divided for clarity such that the table headings of each new table now reflects the adjoining text of the results.

Round 2

Reviewer 1 Report

The authors addressed the reviewer comments, and modified the manuscript when necessary. A final check of the manuscript is recommended.

Reviewer 2 Report

Nicely done!